# mRNA Multipeptide-HLA Class II Immunotherapy for Melanoma

**DOI:** 10.3390/cells14181430

**Published:** 2025-09-12

**Authors:** Apostolos P. Georgopoulos, Lisa M. James, Matthew Sanders

**Affiliations:** 1The HLA Cancer Research Group, Brain Sciences Center, Department of Veterans Affairs Health Care System, Minneapolis VAMC, One Veterans Drive, Minneapolis, MN 55417, USA; lmjames@umn.edu (L.M.J.); sande568@umn.edu (M.S.); 2Department of Neuroscience, University of Minnesota Medical School, Minneapolis, MN 55455, USA; 3Institute for Health Informatics, University of Minnesota Medical School, Minneapolis, MN 55455, USA; 4Department of Psychiatry, University of Minnesota Medical School, Minneapolis, MN 55455, USA

**Keywords:** melanoma, neoantigens, human leukocyte antigen (HLA), Major Histocompatibility Complex (MHC), cancer, immunotherapy

## Abstract

Human Leukocyte Antigen (HLA) Class II (HLA-II) molecules bind peptides of phagocytosed non-self proteins and present them on the cell surface to circulating CD4+ T lymphocytes. A successful binding of the presented peptide with the T cell receptor (TCR) activates the CD4+ T cell, leading to the production of antibodies against the peptide (and the protein of its origin) by the B cell and augmentation of the cytotoxic and memory functions of CD8+ T cells. The first and essential step in this process is the successful formation of a stable peptide-HLA-II complex (pHLA-II), which is achieved when the peptide binds with high affinity to the HLA-II molecule. Such highly antigenic non-self peptides occur in melanoma-associated proteins and could be used as antitumor agents when bound to a matching HLA-II molecule. The objective of this study was to identify such peptides from 15 melanoma-associated proteins. We determined in silico the predicted binding affinity (IC_50_) of all pHLA-II pairs between 192 common HLA-II molecules and all possible linear 15-amino acid (15-*mer*) peptides (epitopes) of 15 known melanoma-associated antigens (N = 3466 epitopes) for a total of 192 × 3466 = 665,472 determinations. From this set, we identified epitopes with strong antigenicity (predicted best binding affinity [PBBA] IC_50_ < 50 nM). Of a total of 665,472 pHLA-II tested, 5941 (0.89%) showed strong PBBA, stemming from 117 HLA-II alleles and 679 distinct epitopes. This set of 5941 pHLA-II pairs with predicted high antigenicity possesses the requisite information for devising multipeptide vaccines with those epitopes alone or in combination with the corresponding HLA-II molecules. The results obtained have a major implication for cancer therapy, namely that the administration of subsets of the 679 high antigenicity epitopes above, alone or in combination with their associated HLA-II molecules, would be successful in engaging CD4+ T helper lymphocytes to augment the cytotoxic action and memory of CD8+ T lymphocytes and induce the production of antitumor antibodies by B cells. This therapy would be effective in other solid tumors (in addition to melanoma) and would be enhanced by concomitant immunotherapy with immune checkpoint inhibitors.

## 1. Introduction

### 1.1. Melanoma

Malignant melanoma originates from melanocytes, specialized cells that produce the pigment melanin. Environmental influences (e.g., exposure to ultraviolet radiation), coupled with genetic events such as mutations in regulatory genes, transform typical-functioning melanocytes into melanoma cancer cells [1]. Cutaneous melanoma is the most serious form of skin cancer, accounting for the vast majority of skin cancer deaths [2], but it can occur in other tissues (e.g., mucosal, uveal) [3]. The high mortality rate of melanoma is attributable to conversion from in situ to invasive form and metastases [1]. While there has been a significant increase in global melanoma incidence over the last several decades, improvements in detection and advances in treatment have contributed to a decline in overall mortality [4,5,6]. Still, many invasive melanoma cases result in death, highlighting the need for novel interventions.

### 1.2. Melanoma Cancer Antigens

As melanoma progresses, the impacted cells express specific antigens that can be used for diagnostic purposes and have become targets of immunotherapy [7]. Melanoma antigens include melanocyte differentiation antigens (MDAs), which are exclusively expressed in melanocytes and overexpressed in melanoma, cancer testis antigens (CTAs), which are typically expressed in normal germline cells but can be expressed in many types of cancer, and other proteins, such as S100, which is expressed in melanomas as well as other cancers and disorders. MDAs include Tyrosinase, TRP-1 and TRP-2, gp100 (also called premelanosomal protein; PMEL17), and Melan-A (also called melanoma antigen recognized by T cells; MART-1). CTAs include those of the melanoma-associated antigen (MAGE), B-M antigen-1 (BAGE), G antigen (GAGE), and NY-ESO-1 families. The role of each of these antigens, as well as their potential for diagnostic purposes or therapeutic intervention, has been reviewed elsewhere [7]. Melanoma is prone to a high rate of mutations [8], producing what are now commonly referred to as neoantigens [9,10]. Neoantigens are patient-specific, vary with regard to immunogenicity, and may be heterogeneous both within a tumor and relative to metastatic sites [11,12].

### 1.3. Melanoma Treatment

Treatment for early stages of melanoma typically involves surgical removal of the tumor and surrounding tissue, resulting in high survival rates [13]; however, more advanced stages of melanoma require additional interventions, including targeted therapies and/or immunotherapy. More specifically, with respect to the latter, the development of immune checkpoint inhibitors (ICIs), T cell therapies, and other melanoma immunotherapies has significantly improved outcomes for some patients, albeit with several limitations, including immune-related adverse effects, immunotherapy resistance, re-occurrence of the tumor, and time-consuming and costly manufacturing of treatment components [14,15,16]. Combination immunotherapies hold promise. Indeed, greater than 50% five-year survival has been documented in patients with metastatic melanoma treated with combined ICIs [17]; however, the lack of sustained benefit in a large portion of cases highlights the need for further development of melanoma immunotherapy. Recent technological advances in vaccine development have ushered in a new era of cancer treatment that holds promise for melanoma and other malignancies.

#### Melanoma Vaccines

Vaccination, generally speaking, is aimed at generating a durable immune response against specific antigens through human leukocyte antigen (HLA)-mediated T cell and B cell activation. The same principles apply to melanoma vaccines, which target either tumor-specific antigens (TSAs) or tumor-associated antigens (TAAs), including MDAs and CTAs [11]. Numerous melanoma vaccine formulations and platforms have been developed over the last decades, including the use of peptides, nucleic acids, viral vectors, dendritic and B cells, with recent substantial progress and development centered around synthetic peptides or nucleic-acid encapsulated liquid nanoparticles [12]. Historically, cancer vaccines for malignant melanoma have demonstrated limited efficacy [18]. However, numerous promising synthetic peptide or nucleic acid-based melanoma vaccines are under investigation in clinical trials [19,20,21]. A critical issue with regard to the effectiveness of nucleic-acid-based vaccines is the selection of antigens that are immunogenic [22], a feature that depends on the patient’s leukocyte antigen (HLA) genetic makeup.

### 1.4. Human Leukocyte Antigen (HLA)

HLA, also commonly referred to as Major Histocompatibility Complex (MHC), comprises cell-surface glycoproteins that play a critical role in human immune surveillance and response. Their primary function is presentation of antigens (peptides) to lymphocytes in order to identify and facilitate immune system activation against foreign antigens and cancers [23]. There are two main classes of HLA (Class I and Class II) that differ in terms of their structure and antigen presentation pathways but share a similar goal of host protection [24]. Briefly, HLA-I molecules of the three classical HLA-I genes (A, B, C) are expressed on the surface of all nucleated cells, including tumors, and are instrumental in the presentation of short (8–10 amino acid residues, mostly 9-*mer* [25]) endogenous antigen peptides to cytotoxic CD8+ T cells. HLA-II molecules of the three classical HLA-II genes (*DPB1*, *DQB1*, *DRB1*) are expressed on professional antigen presenting cells (APC), including macrophages, B cells, and dendritic cells (DC), and are critical for presentation of larger (13–22 amino acid residues, mostly 15-*mer* [26] exogenous antigens to CD4+ helper T cells which stimulate production of antibodies and augment the functions of CD8+ T cells. Subsets of CD8+ and CD4+ T cells acquire immunological memory following activation for faster and effective activation in future encounters with the antigen to which they were exposed. The HLA region is the most highly polymorphic in the human genome [27], contributing to population-level protection against diverse pathogens; however, each individual carries only 12 classical HLA alleles, including 2 from each Class I gene (*HLA-A*, *HLA-B*, and *HLA-C*) and 2 from each Class II gene (*HLA-DPB1*, *HLA-DQB1*, *HLA-DRB1*), that code for the HLA molecules each individual possesses. Variability across HLA polymorphisms is primarily located in the binding groove [28], the amino acid sequence and structure of which determine which antigens can bind with sufficiently high affinity to confer stability to the peptide-HLA (pHLA) complex for an effective presentation to, and activation of, CD8+ and CD4+ T cells [29]. Single amino acid substitutions alter the binding groove [30,31,32], thereby influencing the repertoire of antigen peptides that can bind with HLA for presentation to T cells [33].

#### 1.4.1. Peptide-HLA Binding

HLA molecules of both classes possess very high specificity and degeneracy in peptide binding [30,34]. With respect to the former, a specific HLA molecule can bind with high affinity to only a few non-self antigens among a large number available, and can distinguish peptide antigens by even single amino acid substitutions [30,31,32]; and with respect to the latter, a particular HLA molecule can bind a large number of peptides.

#### 1.4.2. Binding of pHLA to T Cell Receptor (TCR)

The pHLA complex on the surface of the cell presents its peptide to circulating T cells. Although the term TCR is used for both CD8+ and CD4+ T cells, it actually differs in its biophysics between the two cases. Nevertheless, in both cases, peptide-TCR binding is characterized by very high specificity, sensitivity, and degeneracy. With respect to specificity, even a single amino acid residue substitution in the peptide can make a big difference in the probability of its binding to TCR, for both pHLA-I and pHLA-II molecules. With respect to sensitivity, a CD8+ T cell can be activated by a single pHLA-I complex [35], and a CD4+ T cell can be activated by a single pHLA-II complex [36]. And with respect to degeneracy, a single TCR of a CD8+ or CD4+ cell can be engaged by many pHLA complexes.

### 1.5. HLA Expression in Melanoma Cells

#### 1.5.1. General

Both HLA Class I and Class II can be expressed on melanoma cells [37,38]; HLA-I expression occurs in >90% of melanoma cells, whereas HLA-II expression occurs in 50–60% of the cells [37]. Higher expression of HLA (e.g., Class II HLA-DR) has been linked with better prognosis [38,39,40,41], particularly in association with ICI immunotherapy [41]. Secretion of interferon-γ by cytotoxic CD8+ T lymphocytes increases HLA expression in melanoma, thereby increasing antitumor immunogenicity [11]; however, as with other types of cancer, alterations in HLA expression are common in melanoma [42]. Downregulation or loss of HLA is generally associated with worse outcomes [43].

#### 1.5.2. Influence of HLA Melanoma Expression on Outcomes of Immunotherapy

Mounting evidence has documented the influence of HLA on immunotherapy outcomes. In particular, Class II HLA-DR expression has been associated with better outcomes of immune checkpoint blockade immunotherapy, including longer overall survival and progression-free survival for patients with melanoma [41]. However, the influence of HLA on treatment outcomes is not limited to overall HLA expression but also rests on the specific HLA that an individual carries. For example, alleles of the Class I HLA-B44 supertype have been associated with better immune checkpoint blockade immunotherapy outcomes, whereas alleles of the HLA-B62 supertype (particularly HLA-B*15:01) are associated with poor outcomes [44]. Notably, compared to the HLA-B62 supertype, immunogenicity of the HLA-B44 supertype with melanoma antigens is significantly better yet varies across melanoma antigens [45], highlighting the relevance of both the tumor antigens and HLA composition for immunotherapy outcomes. Indeed, personalized treatment approaches based on high-affinity antigen peptide-HLA (pHLA) complexes that are highly immunogenic have become a focus of cancer immunotherapy [46,47,48,49].

#### 1.5.3. HLA and Melanoma Vaccines: The Present Study

Melanoma vaccines represent a rapidly developing approach to melanoma immunotherapy aimed at augmenting the ability of a patient’s immune system to attack cancer cells by enhancing antigen-specific immune responses. With this approach, bioinformatics tools can be leveraged to predict pHLA binding affinity and immunogenicity in order to identify candidate epitopes that can be provided to a patient via various vaccine platforms, including peptide-based vaccines and nucleic acid-based vaccines [50,51]. We have previously used an in silico approach to evaluate the immunogenicity of all possible 9-*mer*s of 11 melanoma-associated antigens to 2462 HLA-I molecules [45] and identified nine HLA-I molecules that would be highly immunogenic to peptide epitopes from all 11 antigens. Here, we extend that to HLA-II molecules by evaluating the predicted binding affinity of 192 common HLA-II molecules to all possible linear 15-*mer* epitopes of 15 known melanoma antigens to identify (a) peptide epitopes with strong antigenicity (i.e., high binding affinity to HLA-II molecules, and (b) HLA-II molecules with strong binding to many melanoma antigens. The former would be useful in including in multipeptide melanoma vaccines, whereas the latter would form the basis of a new melanoma immunotherapy by inducing their synthesis in tumor, DC, or B cells by delivering their mRNA blueprints, as discussed below.

## 2. Materials and Methods

### 2.1. Melanoma/Cancer Antigens

Fifteen cancer antigens (Table 1) were used based on their known occurrence in melanoma tumors and used in various treatments [7]; of those, six are specific to melanoma and nine are expressed in melanoma as well as in other tumors (labeled [CTA] in Table 1). The amino acid (AA) sequences of the 15 melanoma antigens used were retrieved from the Uniprot Database (https://www.uniprot.org/uniprotkb; accessed 1 April 2025) and are given in Appendix A.

### 2.2. In Silico Determination of Predicted Best Binding Affinities (PBBA) to Melanoma Antigens

We estimated in silico the predicted best binding affinity of the HLA alleles above to the 15 melanoma/cancer antigens above (Table 1). Predicted binding affinities were obtained for antigen epitopes using the Immune Epitope Database (IEDB; http://tools.iedb.org/mhci/, accessed on 4 September 2025) NetMHCpan (ver. 4.1 BA) tool [52]. We tested all 232 alleles of *DPB1*, *DQB1*, and *DRB1HLA-II* genes published in the Appendix A of Hurley et al. [53]. Of those, 192 alleles could be modeled by the software tool above, and the results were analyzed as follows. We used the sliding window approach [54,55] to test exhaustively all possible linear 15-*mer* epitopes of the 15 antigens analyzed (Table 1); the epitope length of 15 amino acids is optimal for HLA-II molecule binding [26,56,57]. The method is illustrated in Figure 1 for PMEL17. For each pair of peptide-HLA molecules (pHLA-II) tested, this tool gave, as an output, the IC_50_ of the predicted binding affinity; the smaller the IC_50_, the stronger the binding affinity. The predicted best binding affinity (PBBA) for each pHLA-II pair was the minimum IC_50_ value of all epitopes tested for the pair. An IC_50_ value of IC_50_ < 50 nm is regarded as strong [58]; we called PBBAs < 50 nM “hits”. Given a protein of *N* amino acid length and an epitope length of 15 AA, there were *N-15*+1 PBBAs. The number of epitopes tested for each antigen (across the 192 HLA-II alleles) is given in Table 1.

### 2.3. Statistical Analysis

The IBM-SPSS statistical package (version 30) was used for standard analyses. All *p*-values reported are two-sided.

## 3. Results

### 3.1. Predicted Antigenicity of Melanoma-Associated Epitopes

#### 3.1.1. Peptide Epitopes

We analyzed 15-*mer* epitopes from 15 melanoma-associated antigens (Table 1) and 192 common HLA-II allele molecules (Appendix A). The total number of pHLA-II PBBA determinations was 3466 epitopes × 192 alleles = 665,472 (Table 1). Of those, 5941 (0.89%) were hits (PBBA IC_50_ < 50 nM; Appendix A), indicating strong antigenicity (“antigenic” group), and were retained for further analyses. The frequency of occurrence of these epitopes across the 15 antigens is given in Table 1 and illustrated in Figure 2. It could be that antigens PMEL17, MAGE1, and TRP1 had high numbers of antigenic epitopes, most likely due to their long amino acid sequences (Table 1). In contrast, none of the GAGE group had any strong binders (Table 1).

#### 3.1.2. HLA-II Alleles

The 5941 immunogenic pHLA-II pairs came from combinations of 117 distinct HLA-II alleles and 679 distinct epitopes (the remaining HLA and melanoma antigen epitopes were not strong binding “hits”). All 5941 epitope/allele combinations with strong binding are shown in Appendix A, Appendix A (sorted by allele), and Appendix A (sorted by peptide/epitope). These alleles and the number of antigens of the associated epitopes are shown in Table 2. It can be seen that five alleles (DRB1*01:01, DRB1*01:18, DRB1*01:24, DRB1*01:29, DRB1*10:01) had hits from all 11 antigens that showed strong bindings. The frequency of occurrence of the 679 distinct epitopes with strong binding is shown in Appendix A and illustrated in Figure 3. Finally, Appendix A shows all pHLA-II pairs arranged by allele, and Appendix A shows the number and percentage of hits across the 117 alleles, ranked from high to low percentage. Finally, the frequencies of those alleles among five ethnic populations are given in Appendix A.

With respect to HLA-II genes, we found the following. (a) There were no DQB1 alleles with strong binding; (b) the proportion of DPB1 alleles did not differ significantly between the antigenic group (23/117 = 0.197) and the original group (41/192 = 0.213) (*p* = 0.72, Z = 0.36, test of two proportions); in contrast, (c) the proportion of DRB1 alleles was highly significantly higher in the antigenic group (94/117 = 0.803) than the original group (116/192 = 0.604) (*p* = 0.0001, Z = 3.81, test of two proportions).

## 4. Discussion

### 4.1. Methodological Considerations

Two aspects of our methodology are noteworthy with respect to peptides and alleles tested. With respect to the former, a key point was the testing of all possible linear 15-*mer* peptides of all 15 melanoma-associated antigens. This procedure lends credibility to the set of peptides identified as strong binders (IC_50_ < 50 nM), eliminating guessing or peptide selection for vaccines based on a general description of antigens as “immunogenic”. And with respect to the latter, testing 192 common alleles [53] provided a wide spectrum of HLA-II representation. The combination of these approaches broadly and robustly covers the interactions between melanoma antigens and HLA-II molecules.

### 4.2. Antigenicity and Immunogenicity

The terms “antigenicity” and “immunogenicity” are being used in the literature frequently interchangeably. Strictly speaking, in the context of antigen processing and presentation by HLA-I/HLA-II molecules, antigenicity refers to the probability and strength of attachment of a specific peptide to a specific HLA molecule to form a stable pHLA complex; the immunogenicity of that pHLA complex refers to the probability and strength of eliciting an immune response by the T cells that the pHLA complex engages. The following points are noteworthy. (a) The key factor for both antigenicity and immunogenicity is the pHLA complex, not the peptides by themselves. (b) High antigenicity (i.e., strong binding) is a prerequisite for high immunogenicity. (c) The next step in an immune response is the binding of the peptide presented by the pHLA complex to the T cell receptor (TCR). Although the same generic notation (TCR) is used for both CD8+ and CD4+ molecules, the biophysics are different in the two cases, since two different, T cell-specific, coreceptors are involved, namely CD8 for the pHLA-I-specific CD8+ T cells and CD4 for the pHLA-II-specific CD4+ T cells. Both coreceptors increase the stability of the pHLA complex and facilitate the ultimate activation of the corresponding T cells for an effective discharge of their effector actions (cytotoxicity for CD8+ T cells; initiation of antibody production and enhancement in CD8+ T cells’ actions by CD4+ T cells). (d) Importantly, as mentioned above, the peptide-TCR binding is characterized by very high specificity, sensitivity, and degeneracy. With respect to specificity, even a single amino acid residue substitution in the peptide can make a big difference in the probability of its binding to TCR, for both pHLA-I and pHLA-II molecules. With respect to sensitivity, a CD8+ T cell can be activated by a single pHLA-I complex [35], and a CD4+ T cell can be activated by a single pHLA-II complex [36]. And with respect to degeneracy, a single TCR of a CD8+ or CD4+ cell can be engaged by many pHLA complexes. (e) As mentioned above, the strong, high-affinity binding of a peptide to the HLA molecule renders the resulting pHLA complex very stable. This pHLA stability increases the time for which the pHLA complex is presented to T cells and thus has a direct impact on immunogenicity by increasing the chance of pHLA-TCR binding.

In summary, the primary and most important factor for both antigenicity and immunogenicity is the strength of binding affinity of a peptide to an HLA molecule. The determination of this binding affinity in vitro is tedious, time-consuming, and practically unrealistic for testing large combinations of pHLA complexes. For example, to test in vitro 665,472 pHLA-II complexes evaluated in this study would be infeasible. Fortunately, due to major improvements in bioinformatics, the pHLA binding affinity can be estimated in silico with fair certainty [59]. For that purpose, in this study, we used the freely available NetMHCpan tool [52], a standard in the field.

### 4.3. Antigenic pHLA-II Complexes for Melanoma

Here we identified 5941 (out of 665,472 tested, 0.89%) pHLA-II pairs with strong binding affinities, comprising 679 (out of 3466 tested, 19.6%) distinct 15-*mer* peptides and 117 (out of 192 tested, 60.9%) HLA-II molecules. The peptides came from all but the GAGE antigens (Table 1), and the HLA-II alleles came from the *DPB1* and *DRB1* genes. Interestingly, the percentage of peptides from 15 melanoma-associated antigens found here to bind with high affinity (IC_50_ < 50 nM) to HLA-II molecules is comparable to ~0.5% of peptides from the whole human proteome found to bind, with the same high affinity, to HLA-I allele super types [58]. Finally, all but the GAGE antigens comprised peptides that bound strongly to HLA-II (Figure 2), and five HLA-II alleles bound with high affinity to peptides of all of them (DRB1*01:01, DRB1*01:18, DRB1*01:24, DRB1*01:29, DRB1*10:01).

### 4.4. Implications for Multipeptide HLA-II Restricted Melanoma Vaccines

Most melanoma vaccines use HLA-I-restricted peptides, but HLA-II-restricted peptides [60] have also been used [47]. Moreover, some mutated neoantigen peptides used in a melanoma vaccine engaged CD4+ T cells [46], indicating binding to HLA-II. Appendix A gives all 5941 pHLA pairs arranged by allele. The first step in an application is to determine HLA-II alleles carried by the patient. Of those, only the two alleles of the *DPB1* gene and the two alleles of the *DRB1* gene are useful, since no high-affinity binding peptide was found for alleles of the *DQB1* gene. Next, amino acid sequences of peptides with high-affinity binding to the DPB1 and DRB1 alleles of the patient are selected from Appendix A, synthesized (and may be embedded in longer peptide sequences), pulsed ex vivo into dendritic cells (DC) of the patients, and reintroduced to the patient, following the protocol of DC cancer treatment [61]. This treatment will guarantee that the peptides provided will bind with high affinity to the HLA-II molecules that the patient carries for the formation of a stable pHLA-II complex, an essential prerequisite for peptide presentation to circulating CD4+ T cells and initiation of antibody production by B cells. In parallel, mRNA blueprints of the same peptides are introduced into the tumor, leading to the synthesis of the peptides that will be the targets for cytotoxic antibodies. Since this is a peptide-HLA Class II complex, it will engage and activate CD4+ T cells. In addition, activated CD4+ T cells will augment the protective functions of the CD8+ T cells, with respect to both their cytotoxic action and the establishment of memory T-cells [62]. Although the details of this effect are still being investigated, with new insights being uncovered, there is general agreement about this beneficial effect (against tumor growth) of CD4+ → CD8+ T cell interaction.

Whereas the enhancement in CD8+ T cell function by activated CD4+ T cells is relatively well understood [62], the role of antitumor antibodies is unclear, in contrast to their known role in infection. For example, in a viral infection or vaccination of an immunocompetent host, production of antibodies by B cells typically takes 1–3 weeks. Antibodies act by various mechanisms to protect the host by eliminating current infection and preventing reinfection [63,64,65]. The therapeutic and prophylactic effects of such antibodies against pathogens have been well established. In contrast, with respect to cancer, the therapeutic effect of antitumor antibodies is less clear. Circulating antibodies to TAAs have been found in patients with cancer, indicating a humoral immune response to TAAs. This observation is complemented by the development of antibodies to TAA in response to peptide/epitope vaccines for melanoma [66]. However, the effect of such antibodies on tumor reduction/elimination is unclear. Indirect evidence for their positive effect comes from the observation that melanoma tumors expressing HLA-II proteins, HLA-II(+), have better prognosis than HLA-II(─) tumors [38,41], but this differential effect could also be due to the facilitation of CD8+ T cells by activated CD4+ T cells.

### 4.5. Melanoma Immunotherapy Based on the Synthesis of New HLA Molecules

#### 4.5.1. HLA-II Molecules

Cancer treatment with vaccines containing peptides with predicted high-affinity binding to the HLA-II molecules of the patient is possible only if such peptide-HLA combinations exist. For example, of the 192 HLA-II alleles tested in this study, only 117 were found to bind with high affinity to any of the 3466 peptide epitopes tested exhaustively from 15 melanoma antigens. A definite way to ensure the availability of a high-affinity bound pHLA-II pair would be to induce the synthesis of specific HLA-II molecules that would satisfy that condition [67,68]. More specifically, ideal candidates would be the HLA-II molecules found to bind with high affinity to peptides from all 11 TAA, namely DRB1*01:01, DRB1*01:18, DRB1*01:24, DRB1*01:29, and DRB1*10:01 (Table 2). Nucleic acid (mRNA or viral vector-based) blueprints of these molecules and associated peptides would be administered intratumorally to patients with melanoma tumors expressing HLA-II molecules (~60% in primary and ~50% in metastatic tumors [37]). In HLA-II(─) tumors, they would be pulsed ex vivo to the patient’s DC or B cells, according to the respective protocols of cancer treatment [69]. In both cases, the co-administered peptides would form a stable pHLA-II complex, engaging and activating circulating CD4+ T cells for augmentation of CD8+ T cell cytotoxic and other functions (e.g., promoting memory CD8+ T cells), and the production of antitumor antibodies by B cells.

#### 4.5.2. HLA-I Molecules

Essentially the same approach would be followed using HLA-I molecules and associated high-affinity binding peptides, as proposed recently [68]. In fact, we have already identified nine HLA-I molecules (from a set of 2462 HLA-I alleles) that bound with high affinity to 9-*mer* peptides from 11 melanoma-associated antigens (A*02:14, B*07:10, B*35:10, B*40:10, B*40:12, B*44:10, C*07:11, C*07:13, C*07:14) [70]. Nucleic acid (mRNA or viral vector-based) blueprints of these molecules and associated peptides would be injected intratumorally. Since only the α chain is polymorphic in HLA-I molecules, its synthesis would be simpler than that of the more complex HLA-II molecule. Since practically all melanoma tumors express HLA-I molecules, this treatment would be effective.

#### 4.5.3. Tumor Microenvironment

It should be mentioned that the outcome of the aforementioned interventions would depend on the status of the immune (dys)function at the tumor’s microenvironment [71]. For example, if the expression of the HLA-I/HLA-II molecules is suppressed or impaired, epitopes of the peptides/HLA-II molecules introduced to the tumor may not be adequately presented to CD8+ and CD4+ T cells, the function of which might be impaired as well. Therefore, the expected outcome could be quite variable. These considerations lead to an approach that would involve a direct antigen–antibody reaction without the need to involve the patient’s immune system. This approach involves the blood group AB antigens and the antibodies against them, as discussed elsewhere [72].

#### 4.5.4. Challenges

The main concern in this approach of inducing the synthesis of new HLA molecules is the possibility of autoimmunity. Since these are non-self molecules to the patient, they are highly immunogenic, and it is expected that they would be attacked by humoral and cell-mediated mechanisms, in a situation closely resembling the rejection of an HLA-incompatible organ transplant [73]. In fact, this reaction would be beneficial, as it would lead to the rejection of the tumor, similarly to the case of a HLA-incompatible transplant [72]. Now, one would expect autoimmunity to occur if epitopes of the new HLA molecules are shared with proteins in other tissues, in which case, autoimmunity would occur due to shared epitope/molecular mimicry. Interestingly, such reactions have not been reported in HLA-incompatible organ transplantation, but autoimmune reactions have been documented, actually due to autoimmunity to cryptic self-antigens that appear during chronic damage of the transplanted organ. Apparently, the appearance of such new, antigenic self-antigens would depend on the size of the transplanted organ and the duration of chronic rejection (typically months). However, in the case of melanoma (and other solid tumors), these factors would be minimal since the size of the tumor is much smaller than common transplants (e.g., kidney, lung, liver) and its rejection would be faster. In any case, such autoimmune reactions would be treated using standard protocols, as applied in post-transplantation management of autoimmune reactions.

## 5. Limitations

The general limitation of this study is that it is an in silico investigation of predicted peptide-HLA-II binding affinities. This bioinformatics approach allows for screening a large number of pHLA pairs at the expense of the certainty provided by in vitro assessments of binding affinities. However, the methods and bioinformatics tools used to derive those predictions have improved tremendously during the past several years, aided by the concomitant increase in computer power available for large-scale computations, and, as mentioned above, the pHLA binding affinity can be estimated in silico with fair certainty [59]. For example, the results of this study provide new information on predicted binding affinities based on 665,472 in silico estimations, a feat that would have taken a very long time to accomplish by in vitro methods. Similarly, in a comparable study of binding affinity of all 9-*mer* epitopes from 11 melanoma-associated antigens (N = 3123) to 2462 HLA-I molecules [70], we estimated in silico predicted binding affinities of 3123 × 2462 = 7,688,826 pHLA-I pairs, a practically infeasible task to complete in vitro. However, if the focus of the scientific question is on the binding affinity of a particular peptide to a particular HLA molecule, an in vitro assessment should accompany the in silico prediction [74,75,76].

## 6. Conclusions

The first and single most important step in adaptive immunity is the formation of a stable pHLA complex. This involves both a peptide and an HLA molecule. Here, we carried out an exhaustive testing of all 15-*mer* peptides of 15 melanoma-associated antigens to 192 common HLA-II allele proteins, for a total of 665,472 predicted binding affinity estimations. The 5941 pHLA-II pairs found to bind with strong affinity included 679 distinct 15-*mer* peptides from 11/15 antigens and 117/192 HLA-II molecules of the *DRB1* and *DQB1* genes. Specific peptide-HLA-II molecule combinations can be used for the development of effective vaccines against melanoma, where the synthesis of both the peptide and the HLA-II can be accomplished by injecting intratumorally their mRNA blueprints in HLA-II(+) melanoma tumors, or, in HLA-II(−) tumors, pulsing those blueprints to dendritic cells or B cells ex vivo, with further procedures dictated by the standard DC/B cell treatment protocols [61,69]. This treatment, combined with intratumorally injection of mRNA blueprints of high-binding-affinity HLA-I peptides and molecules [68], would be a useful addition to mRNA-based vaccines against melanoma. Obviously, this and other treatments would be more effective when applied to melanoma tumors at an early stage. In this regard, blood biomarkers such as the fatty acid and protein composition of circulating CD81-positive exosomes [77] hold great promise.

Finally, it should be mentioned that neuroinformatics has been a continuously evolving strong force in vaccine development and design. As explicitly expressed in a recent review of this topic, “Immunoinformatics represents a transformative approach to vaccine research, improving clinical trial efficiency and enabling the development of more reliable, flexible, and personalized vaccines. This approach has the potential to significantly enhance global healthcare outcomes by accelerating the vaccine development process and optimizing vaccination strategies.” ([78], Abstract).

## Figures and Tables

**Figure 1 cells-14-01430-f001:**
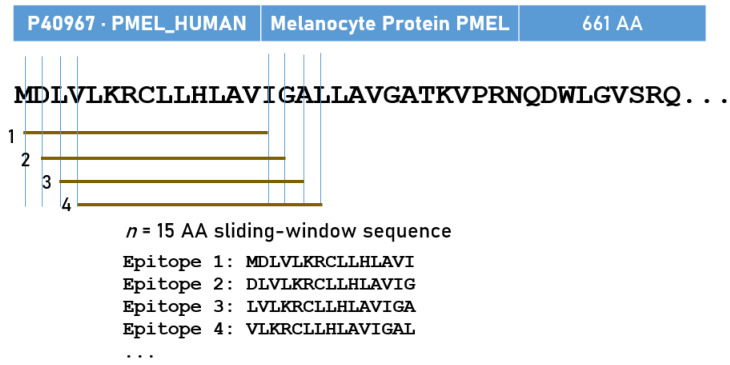
Schematic diagram illustrating the sliding-window method used to evaluate the immunogenicity of all 15-AA linear epitopes of melanoma antigens.

**Figure 2 cells-14-01430-f002:**
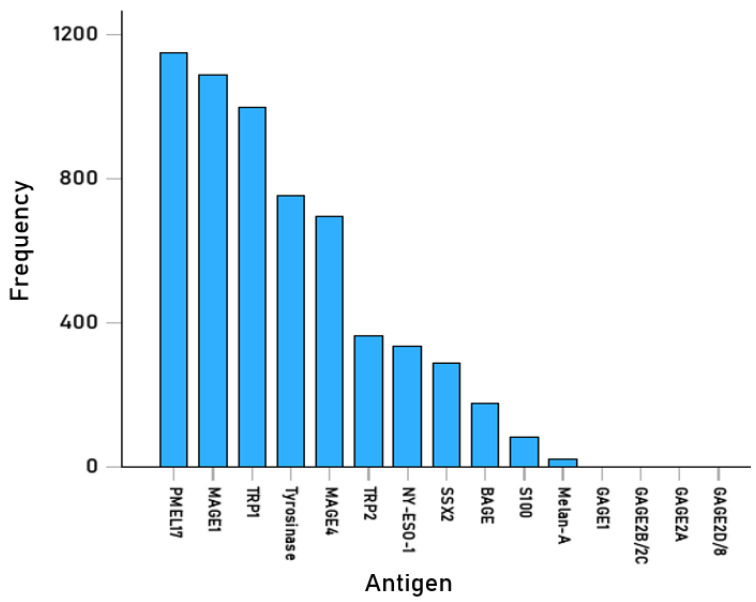
Frequency distribution (per antigen) of 5941 pHLA-II pairs with strong binding affinities.

**Figure 3 cells-14-01430-f003:**
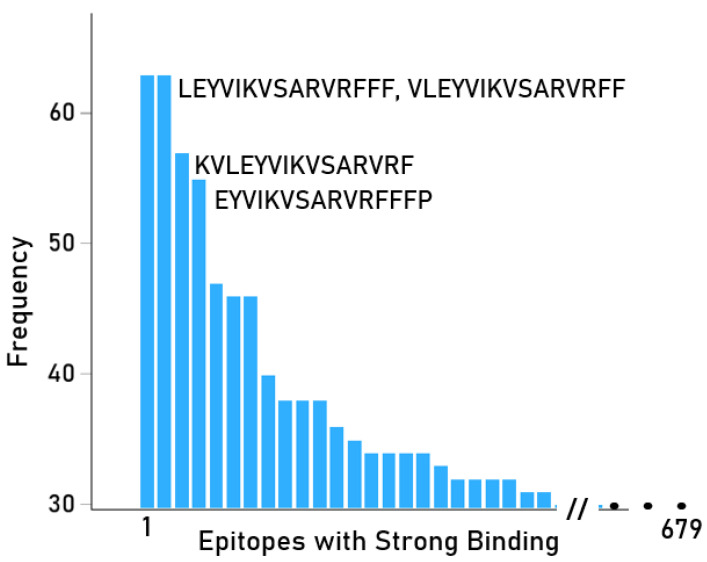
Frequency distribution of 679 distinct epitopes of HLA-II pairs with strong binding affinity. (Appendix A).

**Table 1 cells-14-01430-t001:** The 15 cancer/melanoma-related antigens, the number of epitopes tested, and hits (PBBA < 50 nm). N tested is the number of 15-AA epitopes × 192 HLA-II alleles. The list of antigens is from [7]. [CTA] denotes that the antigen is a member of the Cancer-Testis Antigens family of antigens, also expressed in other tumors.

	Uniprot	Cancer Antigen	N (AA)	N (15-AA)	N Tested	Hits	% Hits
1	O75767	TRP2	237	223	42,816	364	0.85
2	P04271	S100	92	78	14,976	83	0.55
3	P0DTW1	GAGE1 [CTA]	117	103	19,776	0	0.00
4	P14679	Tyrosinase	529	515	98,880	751	0.76
5	P17643	TRP1	537	523	100,416	998	0.99
6	P40967	PMEL17/gp100	661	647	124,224	1151	0.93
7	P43355	MAGE1 [CTA]	309	295	56,640	1087	1.92
8	P43358	MAGE4 [CTA]	317	303	58,176	695	1.19
9	P78358	NY-ESO-1 [CTA]	180	166	31,872	333	1.04
10	Q13066	GAGE2B/2C	116	102	19,584	0	0.00
11	Q13072	BAGE [CTA]	43	29	5568	174	3.13
12	Q16385	SSX2 [CTA]	188	174	33,408	286	0.86
13	Q16655	Melan-A/MART-1	118	104	19,968	19	0.10
14	Q6NT46	GAGE2A [CTA]	116	102	19,584	0	0.00
15	Q9UEU5	GAGE2D/GAGE8 [CTA]	116	102	19,584	0	0.00
Total				3466	665,472	5941	

**Table 2 cells-14-01430-t002:** Counts (N) of melanoma antigens for which an allele was a strong binder. The highest number of antigens per allele is 11, since four antigens did not show any strong PBBA (Table 1). Alleles with strong binding to all 11 antigens are in **bold**. Alleles within the same N are arranged alphabetically.

Allele	N	Allele	N	Allele	N	Allele	N
**DRB1*01:01**	**11**	DRB1*11:62	8	DPB1*46:01	5	DRB1*04:10	2
**DRB1*01:18**	**11**	DRB1*11:65	8	DPB1*47:01	5	DRB1*08:02	2
**DRB1*01:24**	**11**	DRB1*11:74	8	DPB1*72:01	5	DRB1*08:30	2
**DRB1*01:29**	**11**	DRB1*13:01	8	DPB1*81:01	5	DRB1*11:27	2
**DRB1*10:01**	**11**	DRB1*13:05	8	DRB1*04:01	5	DRB1*11:54	2
DRB1*01:11	10	DRB1*13:11	8	DRB1*11:37	5	DRB1*12:03	2
DRB1*01:20	10	DRB1*13:14	8	DRB1*13:07	5	DRB1*14:04	2
DPB1*33:01	9	DRB1*13:50	8	DRB1*15:03	5	DRB1*14:38	2
DPB1*71:01	9	DRB1*14:32	8	DRB1*15:07	5	DPB1*04:02	1
DRB1*07:01	9	DRB1*15:01	8	DRB1*16:05	5	DPB1*105:0	1
DRB1*11:13	9	DRB1*15:06	8	DRB1*14:12	4	DPB1*16:01	1
DRB1*11:14	9	DRB1*16:09	8	DRB1*15:37	4	DPB1*19:01	1
DRB1*11:42	9	DRB1*01:02	7	DPB1*01:01	3	DPB1*34:01	1
DRB1*13:02	9	DRB1*08:04	7	DRB1*04:04	3	DPB1*41:01	1
DRB1*13:23	9	DRB1*11:03	7	DRB1*04:05	3	DPB1*49:01	1
DRB1*13:97	9	DRB1*11:84	7	DRB1*04:72	3	DPB1*55:01	1
DRB1*16:02	9	DRB1*13:96	7	DRB1*08:01	3	DRB1*01:03	1
DRB1*03:11	8	DRB1*15:02	7	DRB1*08:24	3	DRB1*03:15	1
DRB1*09:01	8	DRB1*15:15	7	DRB1*11:11	3	DRB1*04:44	1
DRB1*11:01	8	DRB1*16:01	7	DRB1*11:19	3	DRB1*11:06	1
DRB1*11:02	8	DRB1*04:08	6	DRB1*12:16	3	DRB1*11:07	1
DRB1*11:04	8	DRB1*13:21	6	DRB1*13:61	3	DRB1*12:02	1
DRB1*11:08	8	DRB1*14:06	6	DRB1*13:66	3	DRB1*13:33	1
DRB1*11:10	8	DPB1*02:01	5	DRB1*14:01	3	DRB1*14:02	1
DRB1*11:12	8	DPB1*02:02	5	DRB1*14:54	3	DRB1*14:05	1
DRB1*11:28	8	DPB1*04:01	5	DRB1*16:04	3	DRB1*14:07	1
DRB1*11:29	8	DPB1*126:0	5	DPB1*40:01	2	DRB1*14:23	1
DRB1*11:46	8	DPB1*15:01	5	DRB1*03:01	2		
DRB1*11:49	8	DPB1*23:01	5	DRB1*03:04	2		
DRB1*11:58	8	DPB1*39:01	5	DRB1*03:13	2		

## Data Availability

All data used for analysis are freely available from public databases, as stated in the Materials and Methods section of the manuscript.

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
