# Peer review of "mRNA Multipeptide-HLA Class II Immunotherapy for Melanoma"

_cells, 2025, doi:10.3390/cells14181430_

Round 1
Reviewer 1 Report
Comments and Suggestions for Authors
Georgopoulos et al., Nucleic acid multipeptide-HLA Class II vaccines for melanoma based on in silico predictions of binding affinities of 15 melanoma-associated antigens to 192 HLA Class II molecules. The manuscript presents a comprehensive in silico analysis of melanoma-associated antigens and their binding affinity to HLA Class II alleles, aiming to design multipeptide vaccines. The study is timely and relevant, offering valuable computational insights into melanoma immunotherapy, though several points need clarification and further discussion to strengthen its translational impact.
Comments:
There is no mention about the antigen selection criteria. The paper uses 15 melanoma-associated antigens, but the rationale for choosing these particular antigens (and not others) could be better justified. For example, were they selected based on prevalence, clinical relevance, or prior immunotherapy trials?
Population-level HLA coverage. Although 192 HLA-II alleles were tested, the paper does not analyze global or ethnic allele frequencies. Including coverage estimates (e.g., how many individuals worldwide could benefit) would improve the translational relevance.
The limitations section is present but could better highlight practical challenges, such as possible off-target effects, tumor heterogeneity, or immune evasion. Adding this would balance the otherwise optimistic conclusions. Also, a solid statement on the translational aspects of the immunoinformatically predicted vaccines should be placed along with the reputed references.
Author Response
Georgopoulos et al., Nucleic acid multipeptide-HLA Class II vaccines for melanoma based on in silico predictions of binding affinities of 15 melanoma-associated antigens to 192 HLA Class II molecules. The manuscript presents a comprehensive in silico analysis of melanoma-associated antigens and their binding affinity to HLA Class II alleles, aiming to design multipeptide vaccines. The study is timely and relevant, offering valuable computational insights into melanoma immunotherapy, though several points need clarification and further discussion to strengthen its translational impact.
Comments:
There is no mention about the antigen selection criteria. The paper uses 15 melanoma-associated antigens, but the rationale for choosing these particular antigens (and not others) could be better justified. For example, were they selected based on prevalence, clinical relevance, or prior immunotherapy trials?
REPLY: The 15 antigens used are those described by Pitcovski et al. [1] as commonly occurring in melanoma. Moreover, several of those antigens have been used in the designs of vaccines against melanoma (e.g. NY-ESO-1, gp100, MAGE1, etc.). We now make that point clear in this revision.
Population-level HLA coverage. Although 192 HLA-II alleles were tested, the paper does not analyze global or ethnic allele frequencies. Including coverage estimates (e.g., how many individuals worldwide could benefit) would improve the translational relevance.
REPLY: We now provide the world-wide ethnic population coverage of the alleles with strong epitope binding.
The limitations section is present but could better highlight practical challenges, such as possible off-target effects, tumor heterogeneity, or immune evasion. Adding this would balance the otherwise optimistic conclusions. Also, a solid statement on the translational aspects of the immunoinformatically predicted vaccines should be placed along with the reputed references.
REPLY: We have made these points in this revision.
Reference
- Pitcovski, J.; Shahar, E.; Aizenshtein, E.; Gorodetsky, R. Melanoma antigens and related immunological markers. Crit Rev Oncol Hematol 2017, 115, 36-49, doi: 10.1016/j.critrevonc.2017.05.001.

Reviewer 2 Report
Comments and Suggestions for Authors
The authors performed a study regarding nucleic acid multipeptide-HLA Class II vaccines for melanoma. The study is of interest, however, minor changes are needed.
- In the Methods you should specify better why you have investigated 192 common HLA-II alleles listed in Table S2.
- Improve Figure 1
- While tha algoritms are well-established, the study depends on computational predictions without sufficient experimental/traslational validation. The lack of large-scale in vitro or in vivo confirmation limits the strength of the conclusions. Therefore explain how your results can be validated in the daily clinical practice.
- Variations in antigen expression levels, immune evasion and tumor microenvironment factors are not fully addressed
- Did you evaluate the effect of autoimmunity?
- Please add some sentence about exosomes that in melanoma can be associated with immune activity. In this regard, please red this article and add it in the reference: The Fatty Acid and Protein Profiles of Circulating CD81-Positive Small Extracellular Vesicles Are Associated with Disease Stage in Melanoma Patients. Cancers (Basel). 2021 Aug 18;13(16):4157. doi: 10.3390/cancers13164157. PMID: 34439311; PMCID: PMC8392159.
Author Response
The authors performed a study regarding nucleic acid multipeptide-HLA Class II vaccines for melanoma. The study is of interest, however, minor changes are needed.
- In the Methods you should specify better why you have investigated 192 common HLA-II alleles listed in Table S2.
REPLY: We do so in this revision. Specifically, we tested all 232 alleles (DPB1, DQB1, DRB1) published in the Supplementary Tables of Hurley et. Al. (Hurley, C.K.; Kempenich, J.; Wadsworth, K.; Sauter, J.; Hofmann, J.A.; Schefzyk, D.; Schmidt, A.H.; Galarza, P.; Cardozo, M.B.R.; Dudkiewicz, M.;, et al. Common, intermediate and well-documented HLA alleles in world populations: CIWD ver-sion 3.0.0. HLA 2020, 95, 516-531, doi: 10.1111/tan.13811.), ref [52] in our paper. Of those, 192 alleles could be modeled by the software tools we used to predict binding affinities [Immune Epitope Database (IEDB; http://tools.iedb.org/mhci/) NetMHCpan (ver. 4.1 BA)].
- Improve Figure 1
REPLY: We did and illustrated explicitly the sliding window method for eaxhaustive testing of antigen epitopes.
- While tha algoritms are well-established, the study depends on computational predictions without sufficient experimental/traslational validation. The lack of large-scale in vitroor in vivo confirmation limits the strength of the conclusions. Therefore explain how your results can be validated in the daily clinical practice.
REPLY: We address this issue explicitly in this revision, pointing the strengths and limitations of the bioinformatics approach and reviewing its useful application in other studies where exhaustive in vitro approaches were infeasible. In addition, we have already pointed out in the Limitations paragraph of the discussion that this is a limitation of the study.
- Variations in antigen expression levels, immune evasion and tumor microenvironment factors are not fully addressed
REPLY: We have discussed these points now in this revision.
- Did you evaluate the effect of autoimmunity?
REPLY: We have discussed this potential adverse effect of our proposed treatment and its treatment (if it appears) in this revision.
- Please add some sentence about exosomes that in melanoma can be associated with immune activity. In this regard, please red this article and add it in the reference: The Fatty Acid and Protein Profiles of Circulating CD81-Positive Small Extracellular Vesicles Are Associated with Disease Stage in Melanoma Patients. Cancers (Basel). 2021 Aug 18;13(16):4157. doi: 10.3390/cancers13164157. PMID: 34439311; PMCID: PMC8392159.
REPLY: We added – thank you.
Reviewer 3 Report
Comments and Suggestions for Authors
The manuscript titled “Nucleic acid multipeptide-HLA Class II vaccines for melanoma based on in silico predictions of binding affinities of 15 melanoma-associated antigens to 192 HLA Class II molecules” by Georgopoulos et al. is a well-structured, timely, and engaging research. This study used computer-based methods to find 679 short protein pieces from 15 melanoma-related proteins that can strongly attach to 192 common immune system molecules (HLA Class II). These results could help create new DNA or RNA vaccines that activate the immune system to fight melanoma and possibly other types of cancer. I suggest minor revisions to enhance clarity and strengthen the manuscript’s impact before publication.
Comments for Authors
- The title of the manuscript is too length and dense, the authors are advised to concise the title.
- The abstract uses too many complex scientific terms and is hard to follow. Try to make it simpler or organize it better so that more people can understand it easily.
- Did any of the peptides show the ability to bind to many different HLA-II types?
- How were the 15 melanoma-related proteins chosen? Were they selected because they are highly expressed in melanoma or known to trigger immune responses?.
- Are the selected peptides also found in healthy tissues?
- Is this treatment possible since people have many different HLA types?
- The authors should carefully check for grammatical errors and typos.
Author Response
The manuscript titled “Nucleic acid multipeptide-HLA Class II vaccines for melanoma based on in silico predictions of binding affinities of 15 melanoma-associated antigens to 192 HLA Class II molecules” by Georgopoulos et al. is a well-structured, timely, and engaging research. This study used computer-based methods to find 679 short protein pieces from 15 melanoma-related proteins that can strongly attach to 192 common immune system molecules (HLA Class II). These results could help create new DNA or RNA vaccines that activate the immune system to fight melanoma and possibly other types of cancer. I suggest minor revisions to enhance clarity and strengthen the manuscript’s impact before publication.
Comments for Authors
- The title of the manuscript is too length and dense, the authors are advised to concise the title.
REPLY: We have condensed the title.
- The abstract uses too many complex scientific terms and is hard to follow. Try to make it simpler or organize it better so that more people can understand it easily.
REPLY: We simplified the abstract.
- Did any of the peptides show the ability to bind to many different HLA-II types?
REPLY: Yes! We have provided that information in Table S7 at the Supplementary Material document and discussed the results in the text.
- How were the 15 melanoma-related proteins chosen? Were they selected because they are highly expressed in melanoma or known to trigger immune responses?.
REPLY: The 15 antigens used are those described by Pitcovski et al. [1] as commonly occurring in melanoma. Moreover, several of those antigens have been used in the designs of vaccines against melanoma (e.g. NY-ESO-1, gp100, MAGE1, etc.). We now make that point clear in this revision.
- Are the selected peptides also found in healthy tissues?
REPLY: We don’t know but we doubt they would occur in high frequencies. As we discuss in the paper, the percentage of strong (peptide) binders across the 192 Class II alleles we studied was 5,941/665,472 (0.89%), a percentage very close to ~0.5% of peptides from the whole (normal) human proteome found to bind with the same high affinity (IC50 < 50 nM) to HLA-I allele super types [2]. To our knowledge, no such estimate for HLA Class II molecules has been published but, given the thorough thymic screening out of non-self peptides, we suspect that most (e.g. >99%) strong binders would have been eliminated, otherwise autoimmunity would have been common. We discuss this issue in this revision.
- Is this treatment possible since people have many different HLA types?
REPLY: Yes, for 2 different reasons/applications. (a) First, given the HLA composition of a particular patient, the mRNA blueprint of peptides (epitopes) with strong binding affinity of those alleles would be given intratumorly as vaccine. This treatment utilizes the existing patient’s HLA makeup for peptide selection, and has been used in clinical trials (see refs. 46, 47 in our paper). And second, (b) irrespective of the patient’s HLA makeup, (a) in HLA-II expressing melanoma tumors, the mRNA blueprint of specific HLA-II molecules with matched strong binding to the introduced peptides will be administered intratumorly, to form a stable pHLA-II complex. In HLA-II negative tumors, to the dendritic and/or B cells could be pulsed with the mRNA blueprint of HLA alleles with known strong binding to specific peptides, the mRNA blueprint of which will injected into the tumor.
- The authors should carefully check for grammatical errors and typos.
REPLY: We did.
References
- Pitcovski, J.; Shahar, E.; Aizenshtein, E.; Gorodetsky, R. Melanoma antigens and related immunological markers. Crit Rev Oncol Hematol 2017, 115, 36-49, doi: 10.1016/j.critrevonc.2017.05.001.
- Istrail, S.; Florea, L.; Halldórsson, B.V.; Kohlbacher, O.; Schwartz, R.S.; Yap, V.B.; Yewdell, J.W.; Hoffman, S.L. Comparative immunopeptidomics of humans and their pathogens. Proc Natl Acad Sci USA 2004, 101, 13268-72, doi: 10.1073/pnas.0404740101.
Round 2
Reviewer 1 Report
Comments and Suggestions for Authors
The authors revised the manuscript by considering the reviewers comments, there the revised manuscript is acceptable for publication in its current form.
Reviewer 2 Report
Comments and Suggestions for Authors
The authors improved the manuscritp and it can be accepted for publication.